# Metastable Oscillatory Modes as a Signature of Entropy Management in the Brain

**DOI:** 10.3390/e26121048

**Published:** 2024-12-03

**Authors:** Marta Xavier, Patrícia Figueiredo, Gustavo Deco, Andrea I. Luppi, Joana Cabral

**Affiliations:** 1Institute for Systems and Robotics (ISR-Lisboa) and Department of Bioengineering, Instituto Superior Técnico, Universidade de Lisboa, 1049-001 Lisbon, Portugal; patricia.figueiredo@tecnico.ulisboa.pt; 2Center for Brain and Cognition, Computational Neuroscience Group, Department of Information and Communication Technologies, Universitat Pompeu Fabra, Roc Boronat 138, 08018 Barcelona, Spain; gustavo.deco@upf.edu; 3Institució Catalana de la Recerca i Estudis Avançats (ICREA), Passeig Lluís Companys 23, 08010 Barcelona, Spain; 4Center for Eudaimonia and Human Flourishing, Linacre College, University of Oxford, Oxford OX3 9BX, UK; al857@cam.ac.uk (A.I.L.); joanacabral@med.uminho.pt (J.C.); 5Department of Psychiatry, University of Oxford, Oxford OX3 7JX, UK; 6St John’s College, University of Cambridge, Cambridge CB2 1TP, UK; 7Life and Health Sciences Research Institute, University of Minho, 4710-057 Braga, Portugal

**Keywords:** brain rhythms, entropy management, Free Energy Principle

## Abstract

Entropy management, central to the Free Energy Principle, requires a process that temporarily shifts brain activity toward states of lower or higher entropy. Metastable synchronization is a process by which a system achieves entropy fluctuations by intermittently transitioning between states of collective order and disorder. Previous work has shown that collective oscillations, similar to those recorded from the brain, emerge spontaneously from weakly stable synchronization in critically coupled oscillator systems. However, direct evidence linking the formation of collective oscillations to entropy fluctuations is lacking. In this short communication, we demonstrate how the emergence of Metastable Oscillatory Modes (MOMs) is directly associated with a temporary reduction in entropy in the ongoing dynamics. We apply Shannon entropy to the distribution of eigenvalues of phase covariance over sliding time windows, capturing the temporal evolution of entropy at the level of the entire dynamical system. By demonstrating how the formation of MOMs impacts a system’s entropy levels, we bridge theoretical works on the physics of coupled oscillators with the FEP framework, supporting the hypothesis that brain rhythms recorded experimentally are a signature of entropy management.

## 1. Introduction

Understanding the dynamic processes that govern brain activity is essential for advancing neuroscientific theories and applications. Despite significant advances in neuroimaging techniques, effectively linking neuroimaging data to theoretical frameworks like the Free Energy Principle (FEP) remains a challenge. FEP fundamentally states that the brain minimizes free energy—a concept that we will explain further in the following section and that includes both thermodynamic and statistical interpretations. In both, minimizing free energy involves various mechanisms, including entropy management [1].

Metastable Oscillatory Modes (MOMs) represent collective rhythms emerging from the transient synchronization and desynchronization of coupled oscillatory systems [2,3]. The term ‘metastable’ refers to a temporary state of equilibrium that persists for a significant period but is not a permanently stable equilibrium state of the system [3,4,5]. In this context, critically coupled systems can synchronize and desynchronize forming short-lived collective oscillations whose power is modulated by constant fluctuations in the synchrony degree [6,7]. Such metastable equilibria have even been demonstrated analytically (i.e., solving mathematical equations) in simple networks of oscillators coupled with time delays [6,7]. While analytic predictions can hardly be obtained for the brain’s complex network topology, it is possible to run numerical simulations in computational models and observe the transitory emergence of such theoretically predicted coherent oscillations, or MOMs [2,8,9].

A mechanistic understanding of the synchronization mechanisms driving the formation of MOMs is crucial for understanding the self-organizing principles through which the brain transitions from disordered to ordered states and back, reflecting its dynamic balance between segregation and integration [3]. MOMs can therefore provide a bridge between brain rhythms, such as those detected in M/EEG [10], and the theoretical constructs of the FEP, by reflecting transient synchronization patterns that are associated with varying entropy states. However, this direct link has not yet been shown.

Here, we propose that MOMs are signatures of the brain’s strategy to minimize free energy by effectively managing entropy, thus contributing to the overall stability and functionality of the brain.

### 1.1. Broad Formulation of the Free Energy Principle

The Free Energy Principle (FEP) asserts that biological systems, like the human brain, function to minimize free energy—a concept originating from physics and thermodynamics. In thermodynamics, free energy represents the difference between a system’s total energy and its entropy, indicating the amount of work extractable from the system. As a thermodynamically open system, the brain continuously exchanges matter and energy with its environment, self-organizing to avoid thermal equilibrium, where processes stagnate. This exchange supports the brain’s structural integrity and operational stability, essential for its survival [11]. By adaptively managing entropy, which reflects the disorder within the system, the brain directs itself toward varying entropy configurations. In doing so, it effectively minimizes free energy, maintaining its functional stability [1].

Traditionally, FEP has been interpreted statistically, borrowing from statistical thermodynamics, where free energy is seen as a measure that combines the accuracy of predictions with the uncertainty (entropy) in the probability distributions of a system. In this context, minimizing free energy involves reducing both prediction error and managing uncertainty in probabilistic models of sensory inputs [1]. While the FEP is often discussed alongside theories such as the Bayesian brain hypothesis, predictive coding, and active inference [12], it is important to recognize its distinct scope. Such related theories primarily focus on the computational processes of how the brain updates predictions based on sensory inputs and errors.

In contrast, FEP operates on a broader conceptual scale. It emphasizes the brain’s capacity to manage entropy and maintain minimal free energy, extending beyond the detailed computational mechanics of prediction adjustments. This positions FEP not just as a theory of neural computation, but as a fundamental framework concerned with how biological systems self-organize and sustain stability through effective entropy management. However, the precise physical mechanisms through which this entropy management is achieved remain an open question.

### 1.2. Synchronization Drives Metastable Oscillatory Modes

Metastable Oscillatory Modes represent collective activity states characterized by synchronized oscillations across sets of brain regions that are stable for finite periods of time, before vanishing with a drastic amplitude decay. This type of metastable oscillatory behavior in neural systems has been extensively characterized and observed in M/EEG studies for several decades [13,14,15].

Synchronization is a phenomenon where oscillating systems, such as clocks or fireflies, spontaneously coordinate their rhythms due to interaction [16]. Synchronization manifests in various natural and technological systems, playing a fundamental role in both biological and physical processes. However, in complex networked systems, such as the brain, synchronized states can be naturally disrupted even in the absence of external influences, i.e., due to the intricate interplay of interactions, particularly in [11,17,18].

Computational models simulating the dynamics of coupled oscillators have demonstrated that these metastable oscillatory states emerge from weakly stable synchronization of locally generated gamma (~40 Hz) oscillations, intricately embedded in the complex topology of the brain structural connectome in the presence of naturally occurring time delays [2,8,9,19,20,21,22,23].

Variations in global properties of the brain’s networked dynamical system, such as the coupling strength or conduction speed, influence its dynamical regime, affecting MOMs’ characteristics. Optimal fit with empirical data from resting-state MEG has been achieved within a critical range of parameters placing the system between fully ordered and disordered regimes [2]. In this critical regime, the system spontaneously switches between states of lower and higher synchrony, generating spontaneous fluctuations in the power of collective rhythms and approximating the power spectrum of empirical MEG recordings. This empirical validation strengthens the argument that MOMs are integral to the brain’s self-organizing capacity, but their contribution for entropy management, and in particular the link with theoretical frameworks like the Free Energy Principle (FEP), remains unaddressed.

## 2. Materials and Methods

### 2.1. Computational Model of Coupled Oscillators

To evaluate the relationship between the formation of MOMs and entropy, we ran simulations using the previously published model of coupled Stuart–Landau (SL) oscillators with the parameters found to optimally fit the MEG power spectrum from resting healthy humans [2]. Stuart–Landau oscillators are in a subcritical regime with respect to the Andronov–Hopf bifurcation, exhibiting damped oscillations in response to perturbations. By coupling together N SL oscillatory units according to the connectome, it is possible to describe the complex behavior of N oscillatory brain units, in which the state of each unit n is dependent not only on its intrinsic dynamics, but also on the input received from coupled units, according to the following equations:(1)$$\frac{dZ_{n}}{dt}=Z_{n}\left[a+i\omega -\left|Z_{n}^{2}\right|\right]+K\sum_{p\neq n}^{N}C_{np}[Z_{p}\left(t-\tau_{np}\right)-Z_{n}\left(t\right)]+\beta \;\eta_{1}+i\beta \;\eta_{2},\;\forall n\in N$$
where $Z_{n}$ is the simulated activity of unit n (complex value), ω is the intrinsic frequency of the oscillators, $C_{np}$ is the connectivity strength between units n and p, K is the global coupling strength, and a is the parameter defining the position in respect to the bifurcation. $\tau_{np}$ is the time delay between units n and p (with $\tau_{np}$ = $D_{np}<\tau >$, in which $D_{np}$ is the distance between units n and p and <τ> the mean conduction delay). The terms $\beta {\;\eta }_{1}$ and $i\beta {\;\eta }_{2}$ represent added uncorrelated white noise to each unit, with real and imaginary components $\eta_{1}$ and $\eta_{2}$, randomly drawn from a Gaussian distribution with zero mean and standard deviation β = 0.001.

Each of the *N* = 90 oscillators represents a brain area from a brain parcellation scheme, including subcortical regions but excluding the cerebellum [24]. Although the intrinsic dynamics of each oscillator is identical, their dynamic behavior differs when embedded in the network due to their unique pattern of structural connections. The connections between oscillators are weighted in proportion to the number of fiber tracts detected using in vivo tractography data from diffusion-weighted MRI in healthy humans [2]. This tractography data allows for the construction of a connectome that defines the coupling matrix, where each entry $C_{np}$ represents the connectivity strength between regions n and p (Figure 1a). The coupling matrix is scaled by the global coupling strength, K, a free parameter of the model, that tunes how strongly the brain areas are coupled in the structural network [2]. The mean fiber length between each pair of regions $D_{np}$ is used to calculate the delay $\tau_{np}$ between coupled regions, assuming a homogeneous conduction speed. In Figure 1b,c we report the matrix and distribution of delays, scaled such that the mean is identical to <τ> = 3 milliseconds [2], used in Figure 2.

The resonant frequency of the Stuart–Landau oscillators is set at ω = 40 Hz to mimic gamma band oscillations generated by neuronal circuits. Given that these 40 Hz oscillations usually decay after ~500 ms, the bifurcation parameter is set to *a* = −5, such that the oscillators are in a subcritical regime with respect to the Hopf bifurcation—the point where the oscillation becomes stable instead of decaying back to a stable fixed point. In other words, the units do not oscillate in the absence of input (they have a stable fixed point) but display oscillations at 40 Hz with decaying amplitude in response to input. All units are constantly perturbed with uncorrelated white noise, generating 40 Hz oscillations with fluctuating amplitude. In this setup, the mean conduction delay <τ> and the global coupling strength K are the only free parameters of the model. Importantly, optimal fit with empirical data from resting-state MEG has been previously achieved [2] for parameters K = 10 and <τ> = 3 ms.

### 2.2. Detection and Characterization of MOMs

MOMs are defined as collective oscillations at frequencies lower than the natural frequency, emerging from intermittent synchronization of oscillatory units in the presence of time delays [2,7,25]. To detect MOMs, the simulated signals underwent band-pass filtering within the four canonical sub-gamma frequency bands: delta (0.5–4 Hz), theta (4–8 Hz), alpha (8–13 Hz), and beta (13–30 Hz). Band-pass filtering was achieved by applying a Fast Fourier Transform (FFT), zeroing out frequencies outside the desired range, and reconstructing the filtered signal using an inverse FFT. Subsequently, the amplitude envelopes for each frequency band were extracted using the absolute value of the Hilbert transform, which provides the instantaneous power of the signal in those frequencies.

MOM occurrences were then identified based on both amplitude and collective synchronization criteria. Specifically, a node was considered to engage in a MOM if its amplitude at a specific time point exceeded five standard deviations above the baseline amplitude for that node and frequency band. The baseline amplitude was defined as the mean amplitude of the signal when no time delays were considered in the simulations. This threshold ensures that only significant and substantial increases in amplitude are considered as MOMs, filtering out minor fluctuations due to noise. Additionally, we introduced a minimum coalition size criterion, whereby at least five nodes must simultaneously exceed the amplitude threshold for an event to be classified as a MOM. This threshold was determined based on an empirical analysis of the coalition size distribution (see Appendix A of the Appendix A), ensuring that the detected MOMs reflect genuine collective synchronization rather than isolated fluctuations (see Appendix A of the Appendix A).

In addition to defining the occurrence of MOMs, the coalition size serves as a continuous and quantitative measure of their dynamics. Specifically, it represents the number of nodes that simultaneously exceed the amplitude threshold. This provides a useful metric to capture the emergence and dissolution of the synchronized coalitions that characterize MOMs, allowing us to track their behavior over time.

### 2.3. Entropy of the System over Time

Estimating the level of entropy of an entire system in a given time interval requires a measure of entropy that takes into account the simulated set of N time series altogether, which is different from summing the entropy of each individual time series. In more detail, it is expected that the entropy of the system is maximal if the system is in a fully disordered state, where each unit behaves in its own independent mode of activity. Conversely, if all N time series vary together in the same way, the dynamics of the entire system can be described by a single mode of activity, as if the system’s energy is ‘trapped’ in a collective mode of activity, and therefore, the entropy is expected to be minimal.

The phase covariance matrix can be used to capture the collective dynamics of a system, as it quantifies the synchronization between pairs of oscillators by examining the covariance of their phase angles. By extracting the instantaneous phase of each oscillator using the Hilbert transform, we can estimate the phase covariance matrix across all brain regions at any given point in time.

To obtain a measure that is sensitive to the collective systemic entropy, we first need to understand how the energy is distributed at the system level. This involves performing an eigendecomposition of the phase covariance matrix. The eigendecomposition of a system of N units evolving over time returns a set of N modes of covariance (or eigenmodes), each with: (i) an eigenvalue, which is directly related to the total variance explained by each mode, and (ii) an eigenvector (of size 1 × N), which determines how the N units are organized relative to one another in each mode (i.e., units with the same sign co-vary together, whereas units with different signs co-vary in opposite directions). Each eigenvalue represents the variance captured by the corresponding eigenmode, reflecting how much of the total variability in the data is explained by that mode [26]. This variance can be interpreted as energy, reflecting the magnitude or intensity in that mode. Thus, a higher eigenvalue indicates that more energy is concentrated in that mode, and vice-versa.

Considering the set of N eigenvalues alone, their distribution reveals information about the system’s collective dynamics. If the distribution of eigenvalues is flat, it means that all N modes of covariance have approximately the same energy, whereas if one eigenmode has markedly more energy than all the others, most of the system’s energy is ‘trapped’ in this dominant mode of activity. Several measures are commonly used to characterize a system’s dynamical properties based on its eigenvalue spectrum, such as the ratio of the largest eigenvalue to the sum of all eigenvalues, or the number of eigenvalues needed to explain 95% of the variance.

The Shannon entropy is a formula to estimate the entropy from a probability distribution of possible outcomes or states. Considering that the spectrum of eigenvalues can be treated as a probability distribution of a system’s energy (i.e., each eigenvalue represents a proportion of the total energy, given by the sum of eigenvalues), the Shannon Entropy equation can be used to estimate the entropy of a system in any given interval of time.

Using this rationale, the Shannon Entropy was applied to the probability distribution of eigenvalues obtained for successive N × N covariance matrices computed in sliding windows of 200 ms with 50% overlap over 40 s of simulations (sampled every 2 milliseconds, corresponding to T = 20,000 time points t). While this concept is similar to spectral entropy, which applies Shannon entropy to the power spectral distribution of a signal [27], and von Neumann entropy, which applies it to the eigenvalues of a system’s density matrix [28], the measure used herein used Shannon entropy applied to the eigenvalues of the system’s phase covariance matrix.

For each time point t, the vector of *N* = 90 eigenvalues was normalized such that the sum is equal to 1, obtaining in this way a probability distribution of eigenvalues *p* of size T × N such that the Shannon Entropy, *H*(t), at each instant of time *t* is calculated as:(2)$$H\left(t\right)=-\sum_{n=1}^{N}p(t,n)\;\log\left(p\left(t,n\right)\right)$$

### 2.4. MOMs and Entropy Analyses

To analyze the relationship between MOMs and system entropy, we focused on coalition size as a continuous measure of MOM activity over time, specifically defined as the sum of the sizes of coalitions across all frequency bands (Delta, Theta, Alpha, Beta). To maintain consistency, the coalition size was averaged over the same time windows used for the calculation of Shannon entropy (200 ms with 50% overlap). We then computed the temporal correlation between coalition size and Shannon entropy throughout the entire simulation. This analysis aimed to determine whether larger coalitions are associated with changes in entropy, thereby supporting the hypothesis that MOMs contribute to entropy management.

Although our primary interest lies in the parameter regime that best fits empirical MEG data (K = 10, ⟨τ⟩ = 3 ms; [2]), we conducted analyses across various combinations of global coupling strength (K) and mean conduction delay (⟨τ⟩) to explore how different dynamic regimes influence the relationship between MOM activity and entropy. Specifically, we tested K values of 0.1, 10, and 50, and ⟨τ⟩ values of 0 ms, 3 ms, and 10 ms.

To ensure robustness, we also assessed the influence of different sliding window sizes (ranging from 200 ms to 600 ms, maintaining a 50% overlap between consecutive windows) on the correlation results. While the strength of the correlations varied slightly with window size, their significance remained consistent across the range tested (see Appendix A).

All simulations and data analyses described in this section were performed using MATLAB 2021b (MathWorks, Natick, MA, USA).

## 3. Results

Our results reveal a negative relationship between the size of the coalitions generating MOMs and entropy. In more detail, Figure 2a shows the first 10 s of the simulated signals in the *N* = 90 oscillators over 40 s. Although all the oscillators have a natural frequency at 40 Hz in the gamma frequency band, given that there are time delays between them (proportional to the fiber lengths connecting each pair of brain areas) collective synchronization is stable at frequencies lower than the frequency of the individual units. To highlight this phenomenon, the simulated signals were filtered below 30 Hz. Because time delays play a critical role in generating collective oscillations at frequencies slower than the natural frequency [2,7,9], in the absence of delays there would be no oscillations at these frequencies and the signals detected would be dominated by white noise. As such, the power at these lower frequencies would not (or very rarely) increase above five standard deviations (STD). Here, the oscillations observed in the filtered signals drastically and simultaneously increase their amplitude above the baseline (>5 STD) and, as demonstrated in previous literature, are driven by synchronization. This property generates strong fluctuations in the power at sub-gamma frequencies, which are highlighted in the shaded areas in Figure 2a. These shaded regions indicate the detection of MOMs, identified by thresholding the amplitude at five standard deviations above baseline, along with a minimum coalition size requirement to capture significant synchronization across nodes. As can be seen, after around 1 s of simulations, a large group of oscillators exhibits a simultaneous increase in power in the alpha (8–13 Hz) frequency range that lasts for a few hundred milliseconds, before decaying again. This indicates the detection of a MOM.

During the occurrence of a MOM, the units organize in a mode of collective order explaining most of the variance of the system. Therefore, the eigenvalue distribution shows a steep decline—reflected by fewer dominant modes of variance and consequently reduced Shannon entropy. This observation is evident in Figure 2b, where we show the eigenvalue spectrum for each time window. As can be seen, moments associated with the occurrence of MOMs align with these steeper eigenvalue histograms. On the other hand, periods of lower synchrony imply the contribution of more modes of covariance, and therefore the distribution of eigenvalues is flatter, increasing the system’s entropy.

The evolution of the Shannon entropy over time, shown in yellow in Figure 2c, reveals itinerant dynamics between states of higher and lower entropy. It can be clearly observed that the entropy is minimized after around 1 s of simulations, before increasing again. During this period of lower entropy, the total size of coalitions is also strongly increased. Indeed, over the 40 s of simulated signals, a significant anti-correlation is detected between the Shannon entropy of the eigenvalue distribution and the total size of coalitions across all frequency bands (Pearson’s correlation coefficient r = −0.6625, *p* < 0.05 after Bonferroni correction), highlighting the link between the formation of and entropy management at the system level.

In Figure 3, we extend our analysis to explore different dynamic regimes by varying coupling strength and mean conduction delay. For regimes with low delays and low coupling strength, both system entropy and total coalition size exhibit only minimal variation. Notably, the coalition sizes in these regimes remain consistently below the threshold required for classification as MOMs, suggesting that insufficient coupling or a lack of significant time delays impedes the emergence of collective synchronization phenomena. It should be noted that the coalition sizes depicted in the time-series of Figure 3 are averaged within sliding time windows (as detailed in the Methods section). This averaging was performed to compute correlations with Shannon entropy, which explains why the coalition size values range between 0 and 1, reflecting averaged contributions. In contrast, regions characterized by moderate to high coupling strength (K = 10, 50) and intermediate to long delays (⟨τ⟩ = 3 ms, 10 ms) exhibit significant correlations (*p* < 0.05, after Bonferroni correction) between coalition size and entropy. These regions are associated with higher metastability, with average coalition sizes (summed across frequency bands) varying between 0 and around 200 units, corresponding to greater fluctuations in Shannon entropy. Importantly, these findings include the parameter region known as the “optimal empirical range”, specifically K = 10 and ⟨τ⟩ = 3 ms, which best fits experimentally observed MEG data [2]. This illustrates how metastable synchronization is a signature of entropy management in a network of coupled oscillators operating on an empirically realistic regime.

## 4. Discussion

### 4.1. MOMs as a Signature of Free Energy Management

Our analysis provides evidence to support the hypothesis that MOMs—and more generally, brain rhythms—play an important role in the brain’s adaptive strategy to maintain a rich dynamic repertoire by effectively managing entropy. The synchronization patterns observed in MOMs appear to support the achievement of different entropy states, corresponding to varying levels of neural complexity. This aligns with the FEP’s claim that biological systems, including the brain, aim to minimize free energy, potentially through entropy management. Hence, our findings supporting MOMs as a signature of entropy management suggest that these can be mediated by a fundamental self-organizing principle through which the brain achieves free energy minimization.

### 4.2. Self-Organized Itinerant Dynamics

The itinerant dynamics between higher and lower entropy states emerges spontaneously from fundamental self-organizing principles, providing a mechanism—grounded in the physics of coupled oscillatory systems—to link the theoretical framework of the Free Energy Principle with the brain rhythms detected with M/EEG. As first described by Yeung and Strogatz in 1999, this itinerant dynamics with non-steady order parameters can emerge even in the absence of noise between symmetrically coupled identical oscillators with homogeneous time delays between them [29]. The presence of heterogeneity in the distribution of time delays—naturally occurring in the complexity of neuronal networks—substantially increases the critical range of parameters where multiple coherent attractors can stabilize for finite intervals of time (in more technical terms, these are referred to as hysteresis loops) [6]. As demonstrated by simulations in brain network models, the heterogeneity introduced by realistic distributions of coupling weights and time delays is sufficient to break the detailed balance and to produce the nonequilibrium dynamics characteristic of real brains [19,20,21,22,23,30,31,32].

While the relation between the emergence of collective oscillatory rhythms and entropy minimization was purely inferred from simulated data in this work, the analysis proposed herein can be applied to E/MEG data from participants in different information-processing tasks, in order to evaluate whether such fluctuations in entropy levels link more directly with brain function.

## 5. Conclusions

In this short communication, we aim to put forward the idea that intermittent synchronization—and the subsequent formation of Metastable Oscillatory Modes (MOMs)—serves as a mechanism by which the brain manages entropy to minimize free energy, consistent with Karl Friston’s Free Energy Principle (FEP). We demonstrate that the size of the coalitions generating MOMs is significantly temporally anti-correlated with system entropy. This supports the hypothesis that MOMs are a signature of the brain’s self-organizing efforts to manage entropy, effectively linking theoretical predictions of coupled oscillators with the FEP framework.

## Figures and Tables

**Figure 1 entropy-26-01048-f001:**

Connectivity strength and conduction delays profile of simulation data, derived from diffusion-weighted MRI from the Human Connectome Project (HCP) public database (AAL-90 parcellation, averaged across 32 HCP healthy subjects), for a global coupling strength K = 10 and mean conduction delay < τ > = 3 milliseconds. (**a**) Matrix of connectivity strength, scaled to K = 10. (**b**) Matrix of conduction delays, scaled for <τ> = 3 milliseconds. Connections for which the connectivity strength is zero are colored in white. (**c**) Distribution of conduction delays, after removing entries for which the connectivity strength is zero.

**Figure 2 entropy-26-01048-f002:**
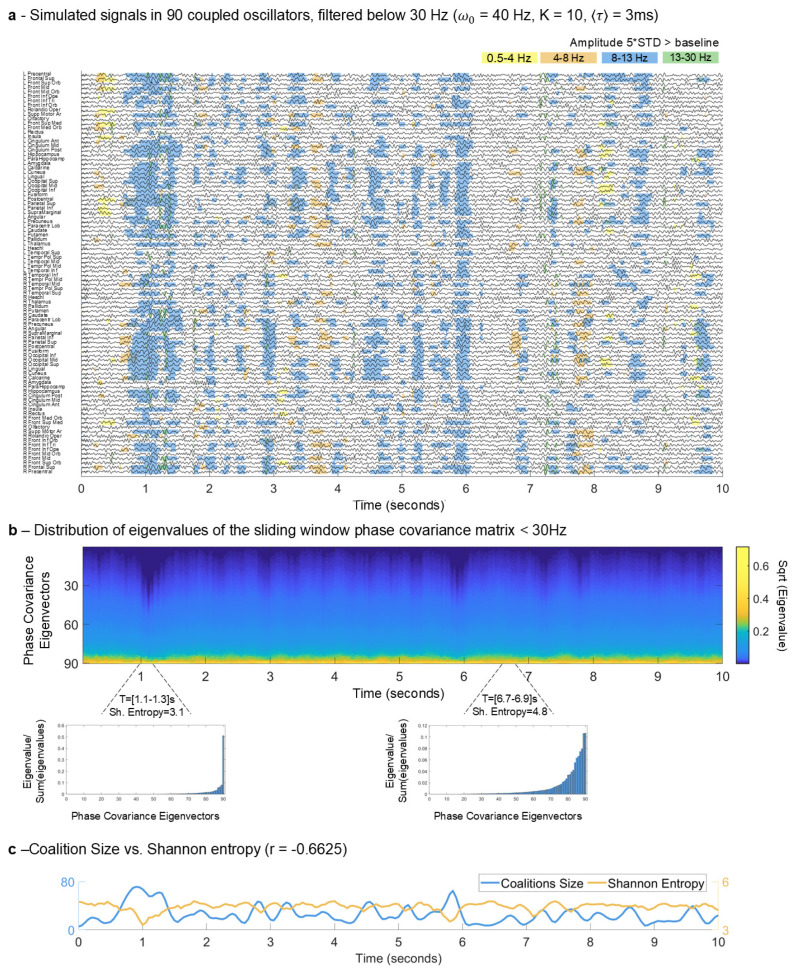
Metastable Oscillatory Modes (MOMs) emerge from cluster synchronization and manage the entropy of the system. (**a**) Simulated brain activity in 90 units plotted over 10 s, each representing a brain area, filtered below 30 Hz to highlight the sub-gamma oscillatory activity typically detected with magnetoencephalography (MEG). Shades indicate MOMs detected in the delta (yellow), theta (orange), alpha (blue), and beta (green) frequency bands. For each frequency band, the amplitude threshold was defined as five standard deviations (STD) of the amplitude—in the same frequency bands—when no delays were considered. The resulting coalitions were then subjected to a size threshold of 5. The simulated signals were obtained for a system of time-delayed oscillators, coupled according to the structural connectome, with the resonant frequency, ω_0_, of all units set to 40 Hz, the conduction speed tuned such that the average delay between units, 〈τ〉, was 3 ms, and the global coupling strength was set to K = 10. The oscillators are sorted in the default order of the parcellation scheme, separating the left and right hemispheres. (**b**) Distribution of eigenvalues of the sliding window (200 ms) phase covariance matrix of the filtered signals shown in (**a**). The occurrence of MOMs (shaded areas of (**a**)) align temporally with a steeper distribution of eigenvalues of the covariance matrices of low-pass filtered signals in the corresponding time windows. (**c**) The total size of coalitions across all frequency bands (blue, left *y*-axis) of the signals shown in (**a**) is significantly anti-correlated (Pearson’s correlation coefficient r = −0.6625, *p* < 0.05 after Bonferroni correction) with the system’s entropy evaluated by the Shannon entropy of the phase covariance eigenvalue distribution (orange, right *y*-axis).

**Figure 3 entropy-26-01048-f003:**
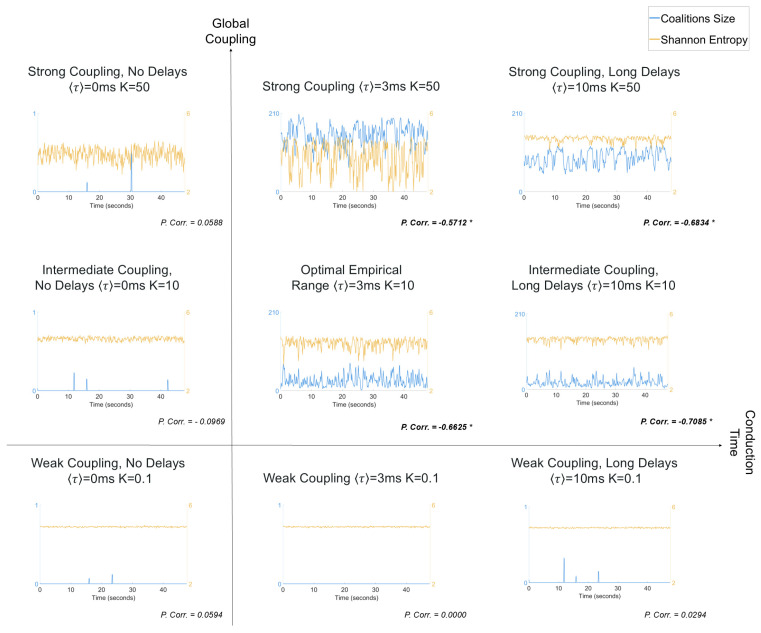
Characterization of entropy dynamics for different values of global coupling strength (K = 0.1, 10, 50) and mean conduction delay (〈τ〉 = 0 ms, 3 ms, 10 ms). Each panel presents the time-series of the total size of coalitions across all frequency bands (Delta, Theta, Alpha, Beta) (blue, left *y*-axis) and system entropy (orange, right *y*-axis). Entropy values were computed and coalition sizes averaged over a sliding time-window (200 ms)—explaining why the coalition size values sometimes range between 0 and 1. The Pearson’s correlation (P. Corr) between coalition size and system entropy are reported below each plot. Significant correlation values (*p* < 0.05, corrected for multiple comparisons with Bonferroni correction) are highlighted in bold and with an asterisk.

## Data Availability

The normative connectomes were computed from Human Connectome Project (HCP) [33] data and included as part of the leadDBS toolbox [34] (https://www.lead-dbs.org/, accessed 15 May 2021). The matrices computed from the normative connectomes used for simulations, together with the codes used for the Hopf simulations, are publicly available at: https://github.com/fcast7/Hopf_Delay_Toolbox, accessed 30 April 2024. Simulated data are available from the corresponding author on reasonable request. The code used to generate the specific results and figures of this study is publicly available at: https://github.com/martaxavier/Hopf_MOMs_Entropy.git, accessed 1 October 2024.

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
