# Peer review of "Metastable Oscillatory Modes as a Signature of Entropy Management in the Brain"

_entropy, 2024, doi:10.3390/e26121048_

Round 1

Reviewer 1 Report

Comments and Suggestions for Authors

he authors use 90 oscillators, representing 90 anatomically parcellated brain areas, to simulate brain activity (metastable oscillatory modes (MOM)) coupled with average conduction delays of 3 ms.  The authors show that “increased power at sub-gamma frequencies during MOM occurrences is significantly anti-correlated with system entropy, supporting their hypothesis that MOMs represent the brain’s efforts to minimize entropy.”

The manuscript is well written for a Short Communication. My only concern is the locations of 90 brain areas represented by oscillators. These are adjacent areas, however, many brain areas are surrounded by several adjacent areas. Are the oscillators connected in the order shown in Figure 1. What is the basis of these connections since the anatomically connected areas do not also share a high level of connectivity. What makes one area different from another since they all have a single oscillator? For example, an oscillator in the caudate nucleus from the thalamus.

Should not the first oscillator represent the thalamus? Also, for the purpose of this paper, naming areas may not be useful.

The fonts used for the brain areas, shown in Figure 1, are very small. The fonts must be made bigger.

Reviewer 2 Report

Comments and Suggestions for Authors

In this manuscript, Xavier M and colleagues aim at assessing whether the collective oscillatory modes, which emerge in the spontaneous brain dynamics, can support the hypothesis that neural systems minimize the free energy principle (FEP) by minimizing the entropy of the system. Although the findings reported here seem to support such claim, I think that the Methods section needs to be expanded significantly. For example, the results clearly show that the average amplitude envelope is associated with a reduction on entropy, but it is not clear what the mean amplitude envelope captures exactly and how it relates quantitatively to the occurrence of a MOM. In addition, the article leverages on the definition of MOMs, but there is no rigorous quantitative definition of a MOM. Without the necessary methodological details listed below, it is not possible to fully assess the soundness of the study. Moreover, additional analyses should be carried to assess the robustness of such findings.

1)    It would be important to include the equations used to simulate the coupled Stuart-Landau oscillators.

2)    How is the power amplitude envelope defined and computed exactly in the simulated data?

3)    It would be beneficial for the paper if the concept of phase covariance matrix is expanded in the methods section.

4)    How were the frequency ranges defined and how were the signals filtered? Please, give the details of all the filtering steps.

5)    I think that in the Methods section there should be a subsection that quantitatively defines a MOM with its properties. For example, it is not clear how MOMs are associated to the 5std threshold.

6)    References and additional explanations can be very helpful in understanding why the signals would be pure noise in case of no time delays (see line 197-199).

7)    The distribution of time delays should be provided among the results.

8)    Further analyses should be carried to explore the robustness of the anti-correlation between the mean amplitude envelope and Shannon entropy. I believe that the following analyses should be added to this study:

a.     How is the anti-correlation dependent on the global coupling strength?

b.     How is the anti-correlation dependent on the conduction delays?

9)    Can the authors provide simulated scenarios with null distribution of either time delays, or coupling strength (or both) where they show that the relationship between mean amplitude envelope and entropy is not valid anymore?

10) Is the code to reproduce the results in Figure 1 available?

Round 2

Reviewer 2 Report

Comments and Suggestions for Authors

I greatly appreciated the work done by the authors in this revised paper where they added additional analyses (summarized in the new Figures 1 and 3) and provided further details in the text about the simulated model and the detection of MOMs. Now I think that it has a higher scientific quality, however I have still many issues (both technically and conceptually) for which, unfortunately, I cannot recommend the publication of the paper as it is. Below, I leave my comments.

Conceptually, I find it still confusing the association between a MOM, which in the text was defined as collective oscillations emerging from the synchronization of oscillatory units in the presence of time delays, and the criterion used to detect MOMs, which is based on the use of the instantaneous power (inst.power), which by definition is associated to a given node signal. In principle, based on the above definition, if the inst. power of a single node signal is above 5SD of the baseline, then there should be a MOM, independently of whether the other nodes are below such threshold. This suggests that “MOMs” are not strictly related to the idea of synchronized oscillatory units, but a MOM can also emerge in a single oscillatory unit. What I understand from the analyses presented in the paper is that there is an anti-correlation between the averaged inst. Power and the entropy computed on the covariance phases within a certain window (200 ms).

Technically, there are many gaps that should be filled before robustly stating that the inst. Power defining MOMs Is anti-correlated to the entropy:

1)    First, there is no statistical analysis on such anti-correlations in general. Figures 2 and (especially) 3 provide only the measure of the Pearson correlation, but there is no statistics testing the significance of such correlation values, for example in the case when there should not be a MOM and therefore no significant anti-correlation (see Fig.3).

2)    Second, there is no comparison between such correlation values in the two distinct conditions when: 1) there is a MOM, 2) there is no MOM. Since the Pearson corrs were used, authors might consider pooling together all time points pertaining to the two distinct conditions and evaluate the Pearson corr coefficients. Importantly, there should be a test of significance on the correlation values when there are no MOMs to show that such anti-correlations are not statistically relevant anymore.

3)    Moreover, the computation of the entropy in a window and its comparison to an instantaneous quantity such as the amplitude is not quite elegant and seems to mix different measures (pairwise vs single channel measures and windowed vs instantaneous).

4)    Finally, based on the results provided in Figure 3, I suspect that the role of connectivity is greater than what was suggested so far. For example, with intermediate values of delay values there is a huge difference in the fluctuations of the averaged inst. Power and entropies between low and high connectivity values. Another point which is still missing in the paper is to provide evidence that there is something special with the intermediate delay values, which would support the emergence of MOMs. For example, it could be added the analysis of manipulating the global connectivity parameter K also in the other two regimes of no and long delays, where MOMs are not expected and prove in those two regimes that the anti-correlations are in general less significant.

Minor point:

1)    In line 70, I don’t think that the paper makes a strong point on suggesting any “optimal” entropy configuration. There is no systematic study of optimality so far.

2)    In line 319 there is no “optimal” range of delay presented so far. For example, if a longer delay parameter was chosen and a higher global coupling parameter was chosen as well, I would expect that other delay ranges show anti-correlation between entropy and inst. Power.

3)    How is the measure of entropy, and the anti-correlations in turn, affected if changing the width of the window from 200 ms to [20-500 ms?

4)    Line 333, what does the “reduction in the complexity” means?

Round 3

Reviewer 2 Report

Comments and Suggestions for Authors

I thank the authors for their efforts in revising this manuscript. I appreciate the more refined definition of MOMs, which now focuses exclusively on collective dynamics. I also value the expanded exploration of the parameter space (Fig. 3) and the additional robustness analyses of the correlation between coalition size and entropy as a function of window size (Fig. S3). I now believe that the paper is technically robust and offers interesting results to the community.